# Lagged effect of temperature and rainfall on malaria incidence in Colombia (2013–2023): An approach with Bayesian spatiotemporal adjustment

**Juan David Gutiérrez** [ID]*

Facultad de Ciencias Médicas y de la Salud, Instituto Masira, Universidad de Santander, Bucaramanga, Santander, Colombia

* jdgutierrez@udes.edu.co

## Abstract

Malaria remains a major public health challenge in Colombia, with a significant increase in cases in recent years. Climate variables—particularly temperature and rainfall—are key drivers of malaria transmission, yet their lagged, non-linear effects across space and time are poorly characterized in the Colombian context. We conducted an ecological, spatiotemporal analysis using weekly malaria case data from 970 municipalities in Colombia (2013–2023) combined with satellite-derived climate data. We applied distributed lag non-linear models (DLNMs) embedded within a Bayesian hierarchical framework using integrated nested Laplace approximation (INLA) to estimate the delayed, non-linear associations between weekly temperature and rainfall and malaria incidence, while accounting for spatial and temporal auto-correlation, forest cover, multidimensional poverty, altitude, population size, and prior case counts. Our results show that malaria risk increases non-linearly with temperature, peaking around 28 °C, with a global exposure of minimum risk (EMR) at 16.43 °C, and significant effects observed at lags of 0–6 weeks. In contrast, lower weekly rainfall was associated with higher malaria risk, with an EMR at 0.85 mm. Sensitivity analyses confirmed the robustness of these findings. These results challenge previous studies about climate-driven malaria risk and highlight accelerated transmission dynamics in Colombia's endemic zones. The identification of specific climate thresholds linked to elevated malaria incidence provides actionable evidence for climate-informed early warning systems and targeted interventions to support malaria elimination efforts in Colombia.

## Author summary

Malaria continues to threaten public health in Colombia, where cases have surged dramatically in recent years. While climate is known to influence

**Data availability statement:** For replication purposes, the code and dataset are available at: https://github.com/juandavidgutier/dlnm_INLA_malaria.

**Funding:** The author(s) received no specific funding for this work.

**Competing interests:** The authors have declared that no competing interests exist.

malaria transmission, the exact timing and shape of these effects—especially how temperature and rainfall affect disease risk with delay—remain unclear in this setting. Using 11 years of municipal-level health and climate data, we applied statistical models that account for both spatial patterns and time lags in climate effects. We found that malaria risk rises with higher temperatures, peaking near 28 °C, and that drier conditions are linked to greater transmission—likely because persistent small water bodies during low rainfall support mosquito breeding without being washed away by heavy rains. Significant temperature effects were concentrated within 0–6 weeks after exposure, indicating faster transmission cycles than previously observed in cooler regions. These insights can help public health authorities design more responsive surveillance and control strategies that anticipate outbreaks based on observed weather patterns, ultimately supporting Colombia's malaria elimination goals.

## 1. Introduction

Malaria, a life-threatening parasitic disease caused by infection with *Plasmodium* protozoa transmitted by infective female *Anopheles* mosquitoes, remains a significant global health challenge despite being both preventable and curable [1]. The disease is primarily caused by five species, with *Plasmodium falciparum* and *Plasmodium vivax* being the most clinically significant and geographically widespread human pathogens [2]. Globally, the epidemiological situation has worsened in recent years, with the WHO reporting an estimated 263 million malaria cases and 597,000 deaths worldwide in 2023, representing an 11 million case increase compared to 2022 [3].

In the Americas, the situation is equally concerning, with countries reporting approximately 505,000 malaria cases in 2023, marking a 5% increase from the previous year, and this upward trend continued into 2024, with over 537,000 cases reported, a 6% increase compared to 2023 [4]. Brazil, Venezuela, and Colombia collectively accounted for nearly 77% of all malaria cases in the Americas region in 2023. Colombia has experienced particularly alarming trends, reporting 105,479 malaria cases in 2023, which represents a 43.4% increase compared to 2022. This surge escalated dramatically in 2024, with 123,740 cases reported, reflecting a 17.3% increase compared to the same period in the previous year [5].

Temperature plays a critical role in malaria transmission dynamics through its influence on both vector and parasite biology. Scientific evidence shows that rising temperatures are associated with increased malaria incidence, with systematic reviews suggesting a moderate positive correlation between mean temperature and malaria cases [6]. According to Mordecai et al., the relationship between temperature and malaria follows a non-linear pattern, with optimal transmission occurring at approximately 25 °C, while transmission efficiency declines significantly above 28 °C [7]. This temperature dependence is biologically mediated, as higher temperatures accelerate parasite development within mosquitoes; for instance, *Plasmodium falciparum* requires only 9 days to complete sporogony at 30 °C compared to 23 days at 20 °C [8].

Recent studies suggest that when temperatures exceed 22.4 °C, malaria transmission increases by 9% to 10% for each additional degree Celsius rise [9]. However, the impact of temperature rise varies geographically, with evidence showing opposing effects on malaria dynamics between lowland and highland regions due to differing baseline climatic conditions [10]. These temperature-malaria relationships have significant implications for climate change adaptation strategies in endemic regions [11].

Rainfall represents an important environmental determinant of malaria transmission dynamics, with numerous studies establishing significant associations between precipitation patterns and malaria incidence [12–14]. Zhou et al. [15] revealed that monthly rainfall is associated with an increase in malaria incidence, but with substantial spatial variation in East African highlands.

The non-linear nature of the association between rainfall and malaria becomes apparent in lowland areas. Matsushita et al. [16] identified an optimal rainfall threshold of 120 mm per month that corresponds to peak malaria transmission risk. Several studies suggest a complex interplay between precipitation and malaria epidemiology, where moderate rainfall creates favorable breeding conditions for *Anopheles* vectors, while excessive rainfall may have washout effects that temporarily reduce transmission intensity [12,14,16].

Climate factors, particularly temperature and precipitation, exhibit delayed effects on malaria transmission dynamics, with various lag periods observed across different epidemiological settings. Previous studies have demonstrated that temperature and rainfall are significantly correlated with malaria prevalence with a lag time of 1–2 months, highlighting the temporal relationship between meteorological conditions and disease incidence [17]. Research in the Amazon region revealed that elevated precipitation and temperature increase the risk of malaria infection in the following two months, with the magnitude of risk influenced by the interaction between these climatic variables and environmental factors [18].

Spatial and temporal autocorrelation significantly influence the accuracy of malaria incidence estimation, as these phenomena create non-random clustering patterns that violate the independence assumption of conventional statistical models. Spatial autocorrelation reveals that malaria cases in Colombia tend to cluster geographically, with neighboring regions exhibiting similar incidence rates due to shared environmental, socioeconomic, and vector-related factors [19]. The spatiotemporal dependency can lead to biased parameter estimates and underestimated standard errors when ignored in regression analyses, potentially resulting in incorrect conclusions about risk factors and control effectiveness, leading to inefficient resource allocation and suboptimal intervention strategies [20,21].

To our knowledge, previous research that simultaneously examined the lagged effects of temperature and rainfall on malaria while utilizing a spatiotemporal statistical framework is scarce [22,23]. This study seeks to fill that methodological gap in Colombia through an ecological analysis conducted from 2013 to 2023. This research provides practical evidence that can help decision-makers and public health officials in Colombia develop new malaria control initiatives. Our findings offer insights for improving disease surveillance and control efforts in areas at risk, particularly by enhancing climate-sensitive approaches. By facilitating targeted interventions and the development of early-warning systems, this work aims to inform strategies for malaria elimination and reduce the risk of outbreaks.

## 2. Methods

We developed an ecological observational study in Colombia, with the municipalities as observation units. We used openly available, anonymized malaria case records, satellite-sensed climate data, and national statistics to estimate the lagged effects of temperature and rainfall on malaria incidence.

### 2.1. Ethical approvals

Ethical authorization for the epidemiological study was granted by the Bioethics Committee at Universidad de Santander (reference code 002, dated February 13th, 2023). The study follows STROBE reporting standards, international protocols designed to improve methodological transparency in health research.

## 2.2. Malaria data

We obtained, on February 21, 2025, an anonymized 11-year retrospective dataset from January 2013 to December 2023, detailing laboratory-confirmed malaria cases across Colombian municipalities. The dataset was retrieved from the SIVIG-ILA platform, the national health surveillance system. Daily cases were grouped by week and municipality of occurrence. Case records with geographic, demographic, or temporal discrepancies were excluded to avoid information bias. Data from municipalities above 1,600 meters in elevation were also omitted, in line with the National Health Institute's established cutoff for malaria transmission risk [24].

## 2.3. Environmental data

**2.3.1. Climate variables.** We obtained daily satellite-sensed data on air temperature at 2 m and rainfall from the ERA5 dataset [25], covering the period from January 2013 to December 2023, with a spatial resolution of 0.10 degrees. We grouped the daily data by week, performed a spatial match between the raster layers and the polygon map, and estimated the weekly averages of temperature and rainfall for each municipality using the raster package in R (version 4.0.3) [26].

**2.3.2. Forest coverage.** We downloaded raster annual layers from 2013 to 2023 of forest coverage from the NASA product MCD12Q1 [27], with a spatial resolution of 500 m. We developed the spatial matching between forest coverage and municipality polygons as mentioned above, and we estimated the annual percentage of area in each municipality with forest coverage.

## 2.4. Socioeconomic data

We obtained data on the Multidimensional Poverty Index (MPI) from the National Department of Statistics [28]. The MPI represents a socioeconomic proxy that quantifies the proportion of households experiencing multi-faceted poverty in Colombian municipalities. It incorporates six measures: access to public services, residential standards, conditions affecting children and young people, educational accessibility, employment prospects, and healthcare availability. Note that the MPI corresponds to a unique measure obtained in 2018, during the last national census.

The yearly population in each municipality was downloaded from the National Department of Statistics [29]. We obtained the minimal altitude in meters of each municipality from the Agustín Codazzi Geographic Institute [30].

## 2.5. Statistical analysis

We implemented a combined statistical approach, leveraging distributed lag non-linear models (DLNMs) with spatiotemporal Bayesian hierarchical modeling to estimate the lagged response of malaria incidence to temperature and rainfall. This approach captures both the temporal and spatial autocorrelation of malaria occurrence and its relationship with weekly meteorological drivers. We developed independent DLNMs for temperature and rainfall, and explored lags from 0 to 8 weeks to capture delays in the exposure-response relationship, covering the time for the response of the vector's population to climate variables, as well as the durations of the intrinsic and extrinsic incubation periods [31].

For the DLNMs, the exposure-response relationship was modelled using natural splines with temperature's knots placed at 20%, 50%, and 80% percentiles of its distribution, and rainfall's knots positioned at 10%, 50%, and 90% percentiles to account for its skewed nature (S1 Fig). The lag-response structure for both variables was modelled using natural splines with knots placed at approximately 1/3 and 2/3 of the maximum lag period (8 weeks), allowing the effect of each exposure at a given time point to vary over subsequent weeks.

The placement and number of knots followed a data-driven strategy guided by the empirical distributions of each exposure (S1 Fig) and by the need to balance flexibility with parsimony in a complex spatiotemporal model. Temperature exhibited an approximately symmetric distribution, so we used a low-dimensional natural spline, with two internal knots (i.e.,

roughly the 1/3 and 2/3 quantiles) to capture moderate, smooth nonlinearity without overfitting. By contrast, rainfall was markedly right-skewed; therefore we placed knots at the 10th, 50th and 90th percentiles to provide additional flexibility in the tails and to allow the model to detect potentially abrupt changes in malaria risk associated with particularly low or very high precipitation.

The modest number of knots (2 for temperature, 3 for rainfall) was chosen to preserve interpretability and to limit model dimensionality given the 8-week lag window and the spatiotemporal structure fitted with the Laplace approximation (see details below). For the lag-response we used natural spline knots at approximately one-third and two-thirds of the maximum lag (8 weeks) so the lag shape can vary smoothly across short and medium delays while keeping a compact parameterization.

This approach enabled the estimation of the overall cumulative effect of each climate variable across the specified lag period, centered around the exposure value associated with the exposure of minimum risk (EMR), while accounting for the non-linear shape of both the exposure-response and lag-response relationships. We restricted the prediction range to the 5th–95th percentiles of temperature and rainfall to minimize the influence of extreme values, which can distort non-linear exposure–response estimates and reduce the stability of the DLNM predictions.

We tested two models— a negative binomial (NB) and a zero-inflated negative binomial (ZINB) — to account for overdispersion in malaria case counts. We selected the best of the two models to estimate the delayed, non-linear associations between weekly temperature, rainfall, and malaria incidence, while accounting for spatial and temporal autocorrelation. The best model was selected based on the Logarithmic Conditional Predictive Ordinate (LCPO), Watanabe–Akaike Information Criterion (WAIC), and Deviance Information Criterion (DIC).

To address unobserved and unmeasured spatial sources of variation, we estimated the spatial random effects using a Besag-York-Mollié (BMY) model. The BYM models are Bayesian hierarchical models that account for spatial auto-correlation by combining a spatially structured conditional autoregressive component with an unstructured random effect to model regional variation in disease rates. Temporal random effects with a cyclic first-order random walk prior were implemented to account for seasonality and unobserved or unmeasured municipal-level factors that change over time.

We implemented the spatiotemporal adjustment using the integrated nested Laplace approximation (INLA), with the R package R-INLA [32]. The log-gamma prior used in our INLA specification places a Gamma distribution on the logarithm of the precision, with parameters 1 and 0.01 corresponding to the shape and rate of the Gamma distribution.

The models take the following general form:

$$\log\left(Y_{m,t}\right) = \alpha + cb\left(c_{i\ mt},\ l\right) + \gamma_{d(m)\ t} + \Psi_{m\ y(t)} + \phi_{m\ y(t)} + \beta_x X_x + \log\left(Y_{t-4}\right)$$

Where $\log\left(Y_{m,t}\right)$ represents malaria cases in the municipality $m$ at week $t$. The intercept is represented by $\alpha$. The cross-basis function of the DLNM, $cb(c_{i\ mt},\ l)$, represents how the effect of the climate variable $c_i$ in the municipality $m$ in the week $t$ on malaria can vary both with the level of exposure and with the time elapsed $l$, since that exposure, using natural splines to model both dimensions. The department-specific $d$ (the administrative level 1) weekly random effects, year-specific spatially unstructured, and structured random effects at the municipality-level are represented by $\gamma_{d(m)t}$, $\Psi_{m\ y(t)}$, and $\phi_{m\ y(t)}$, respectively. $X$ represents the vector of co-variates (i.e., forest coverage, MPI, minimal altitude, and population) with their respective regression coefficients $\beta_x$. $Y_{t-4}$ represents the cumulative lagged four-week malaria case counts (i.e., the cases in the previous month), and it was included to suppress the model's autocorrelation, as suggested by Imai et al. [33].

## 2.6. Sensitivity assessment

We conducted a sensitivity analysis to re-estimate the lagged effects of weekly temperature and rainfall on malaria after changing the parameters of the log-gamma prior to 0.1 and 0.001. Additionally, we included the number of holidays in

each week and the percentage of the municipal area with the presence of illegal mining. We procured remotely sensed data on illegal mining for 2021 (the only available data) from the Colombian Mining Monitoring [34] and developed the spatial matching as mentioned above.

We transformed the values of covariates into standard deviation units to facilitate the convergence of the models. For replication purposes, the code and dataset are available at: https://github.com/juandavidgutier/dlnm_INLA_malaria. The malaria data available on GitHub is a summary that has been carefully scrubbed for privacy. It contains only total counts at the municipal level, with all names, addresses, and any identifiable markers removed. Note that this data was initially anonymized open government information, which was then further processed to remove any personal identifiers.

## 3. Results

A total of 970 municipalities were included in the study. Between 2013 and 2023 were reported 699,860 cases of malaria. Most cases occurred in males (n = 438,112; 62.6%). The top five municipalities with the highest number of cases were Quibdó (Chocó), Tierralta (Córdoba), Inírida (Guainía), Tumaco (Nariño), and Alto Baudó (Chocó).

When we compared the NB and ZINB models, we observed that the NB model showed better performance (Table 1) across all metrics for both temperature and rainfall. For this reason, we used this model for the estimation of the delayed, non-linear associations between weekly temperature, rainfall, and malaria incidence.

Fig 1 presents the diagnostic plots for the temperature and rainfall NB models. Both temperature (Fig 1a) and rainfall (Fig 1b) exhibit consistent diagnostic patterns. The Observed vs Fitted scatterplots display a strong linear association between fitted means and observed counts, indicating that the models capture the central tendency well; however, the dispersion of points increases with larger fitted values and several high-count observations lie above the 1:1 line, indicating systematic under-prediction of extreme counts. The Residuals vs Fitted plots show a clear heteroskedastic pattern (a dense band of small residuals around zero plus a long right tail of positive residuals), meaning residual variance is not constant and large positive deviations remain for a subset of observations. The histograms of Pearson-like residuals are sharply peaked at zero with long right tails, confirming that most observations are well-predicted but a minority of cases produce large positive residuals. Overall, these diagnostics indicate an adequate fit for estimating average effects and spatiotemporal patterns, but a persistent inability to fully reproduce extreme outbreaks.

The overall cumulative association of weekly temperature (i.e., the association centered at the global EMR) and malaria cases showed an increasing trend (Fig 2a). The global EMR was 16.43 °C of temperature. Note that the narrowing of the confidence interval near the global EMR occurs because the model is specifically focused on this exposure level, which represents the global minimum risk point. Meanwhile, the overall cumulative association of weekly rainfall and malaria cases showed a decreasing trend (Fig 2b), with a global EMR of 0.85 mm of precipitation.

**Table 1. Performance metrics of Logarithmic Conditional Predictive Ordinate (LCPO), Watanabe–Akaike Information Criterion (WAIC), and Deviance Information Criterion (DIC) for the negative binomial (NB) and zero-inflated negative binomial (ZINB) models.**

| Metric | NB | ZINB |
|---|---|---|
| Temperature | | |
| LCPO | 207,319.10 | 207,541.00 |
| WAIC | 416,591.10 | 417,142.40 |
| DIC | 417,115.30 | 417,736.70 |
| Rainfall | | |
| LCPO | 207,406.50 | 330,642.10 |
| WAIC | 416,877.00 | 663,308.80 |
| DIC | 417,194.00 | 660,664.40 |

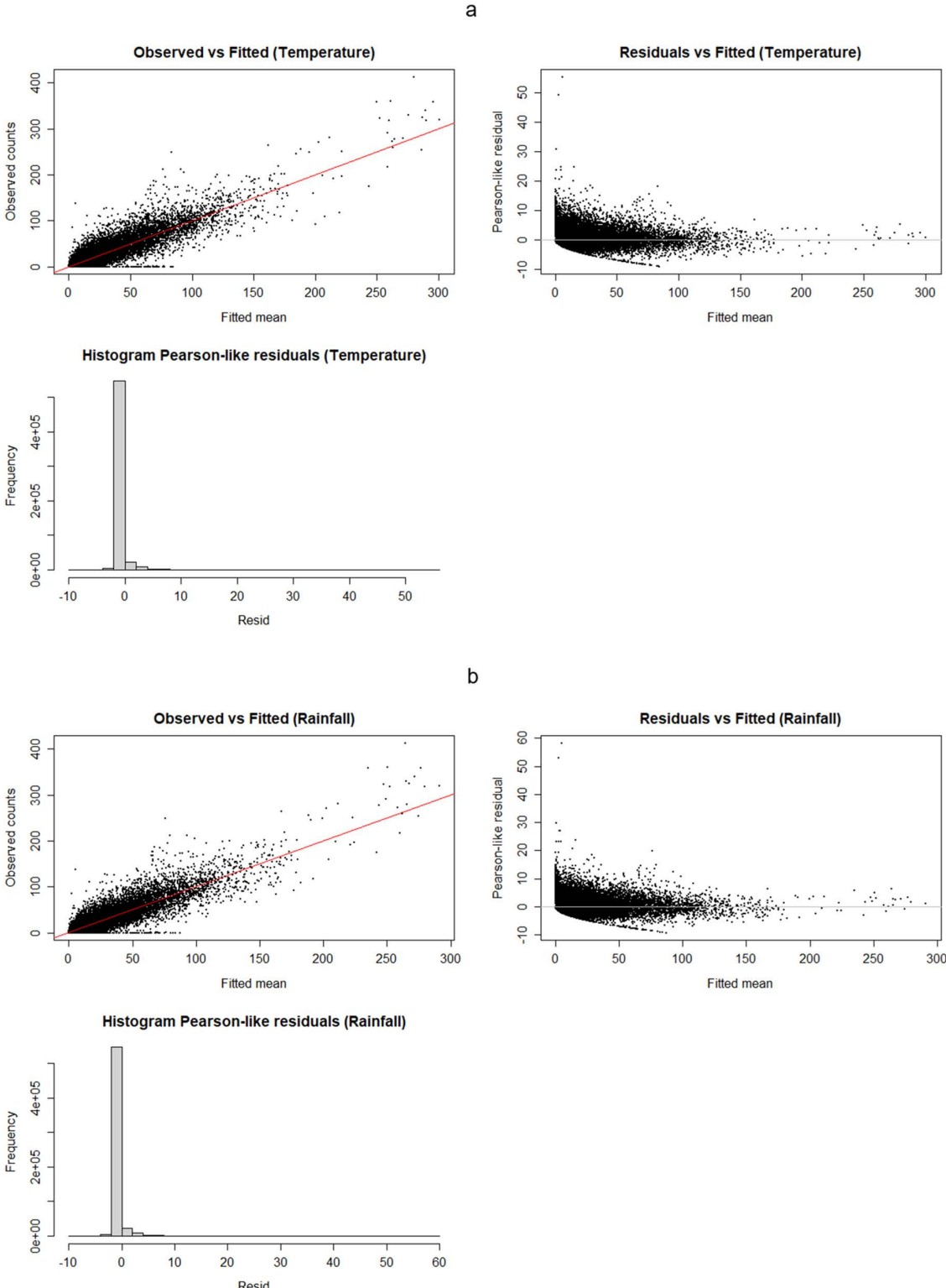

**Fig 1. Diagnostic plots for the negative binomial model for temperature (a) and rainfall (b).** Both panels show Observed vs. Fitted, Residual vs. Fitted, and histograms of Pearson-like residuals for each climate variable.

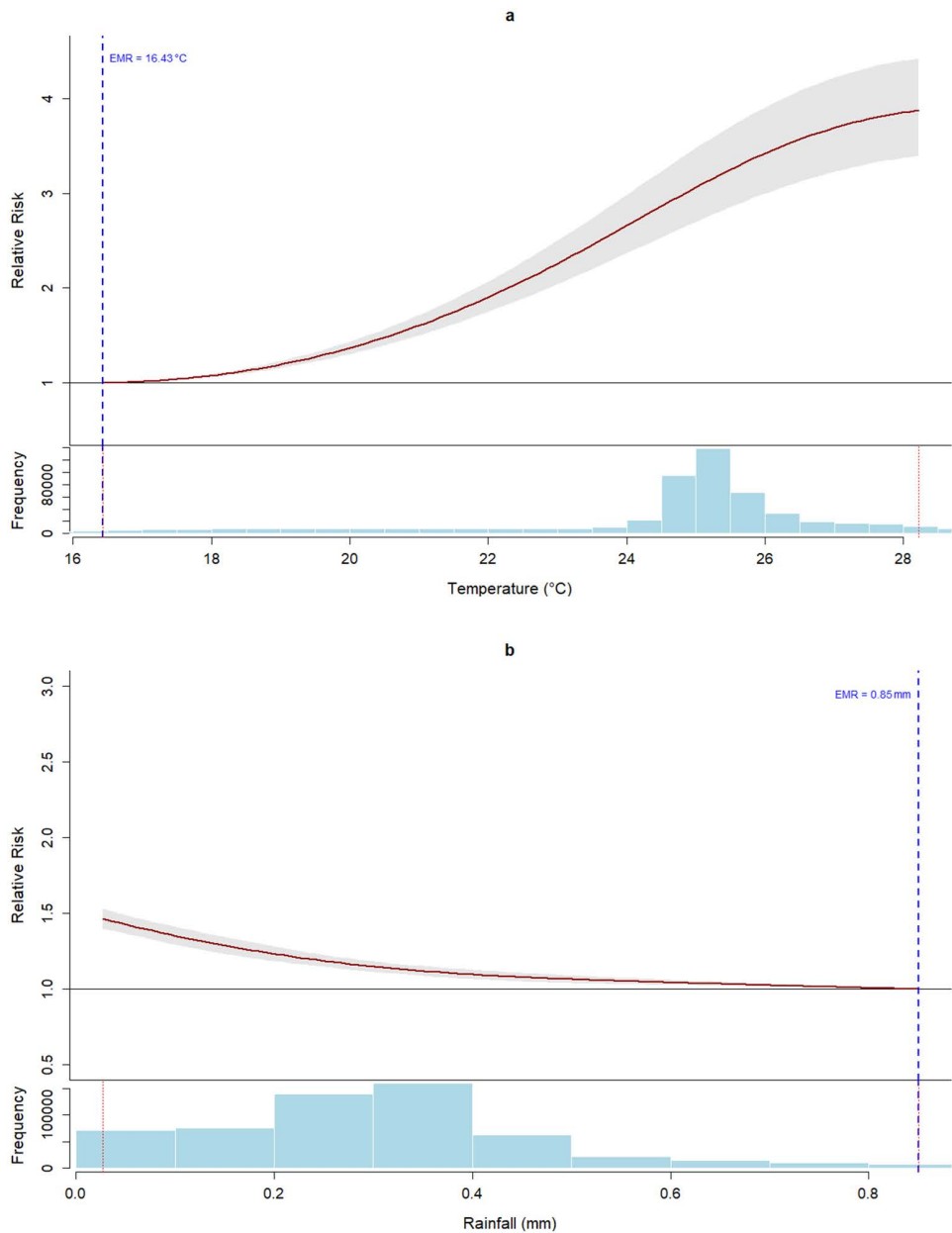

**Fig 2. Overall cumulative association of weekly temperature (a) and rainfall (b) with malaria cases.** The continuous red line corresponds to the point estimate of the relative risk, while the grey band represents the 95% confidence interval. The exposure of minimum risk (EMR) is represented by the dashed vertical blue line. The evaluated interval is shown between the two red dashed vertical lines in the histogram.

The spatiotemporal Bayesian hierarchical model evidenced statistically significant non-linear relationships between climate variables and malaria incidence. A fixed higher temperature corresponds to a larger lag-specific effect on malaria cases, but there is a trend of no significant effects for lags longer than 6 weeks (Fig 3a). Meanwhile, fixed low values of rainfall correspond to a larger lag-specific effect on malaria cases (Fig 3b).

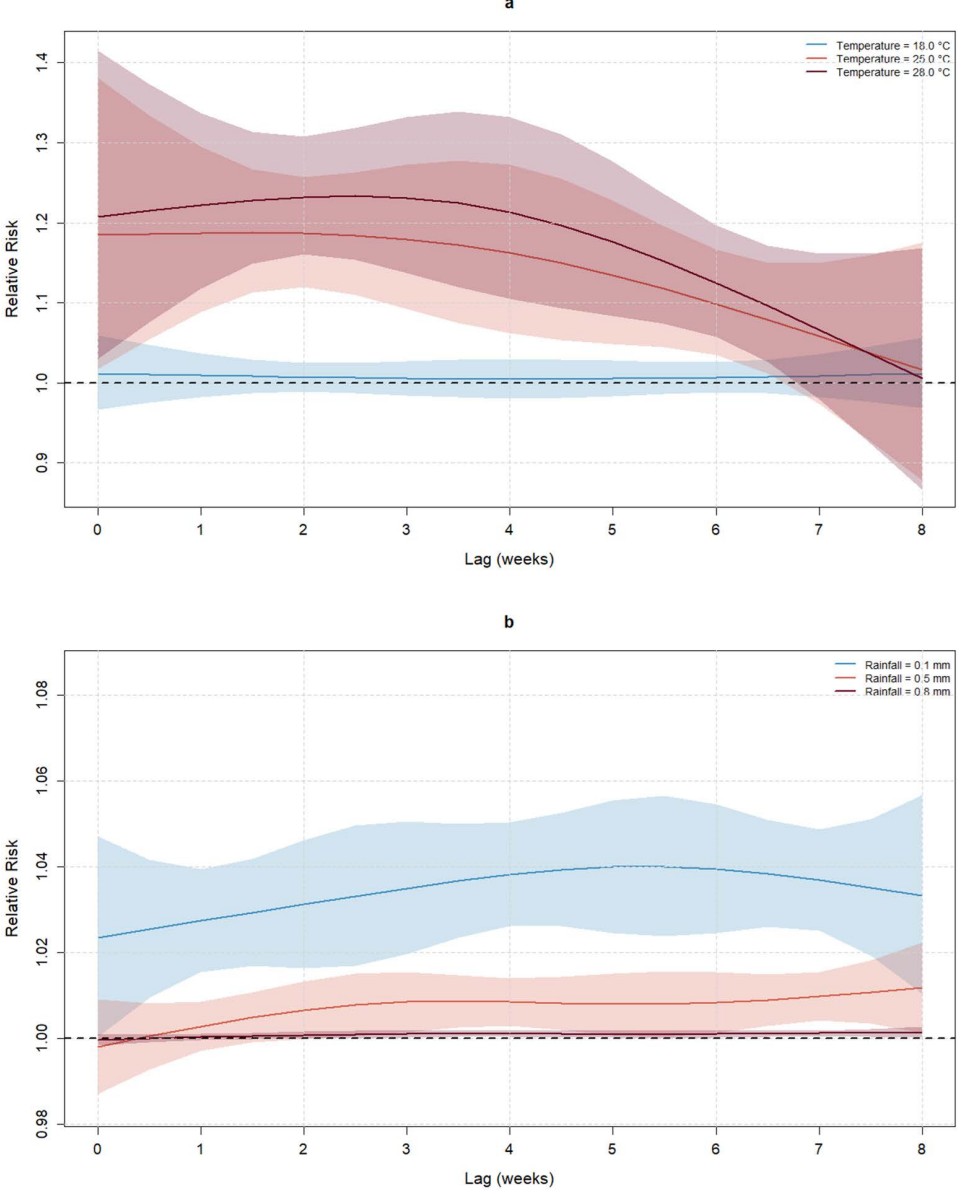

**Fig 3. Lag-specific effect for fixed values of temperature (a) and rainfall (b) on malaria incidence.** The dashed black horizontal line represents the null effect, i.e., where the effect is not different from the effect for the exposure of minimum risk (EMR).

Fixing the lag value revealed an increased effect of temperature on malaria cases, with the magnitude being greater in recent weeks (Fig 4a). In the case of rainfall, when we fixed the value of the lag, there is a trend indicating a negative relationship between rainfall and the incidence of malaria for all the fixed values of lag evaluated. Note that for the fixed value of lag = 2, the effect of rainfall values above 0.5 mm is not significantly different from the null effect (Fig 4b).

Results of the sensitivity assessment, changing the prior parameters and including illegal mining and the number of holidays per week as additional covariates, showed the same values as the main model (Fig 5a and 5b), with the unique exception of a larger effect for the overall cumulative effect of temperate respect to the original estimation (Fig 5a top panel).

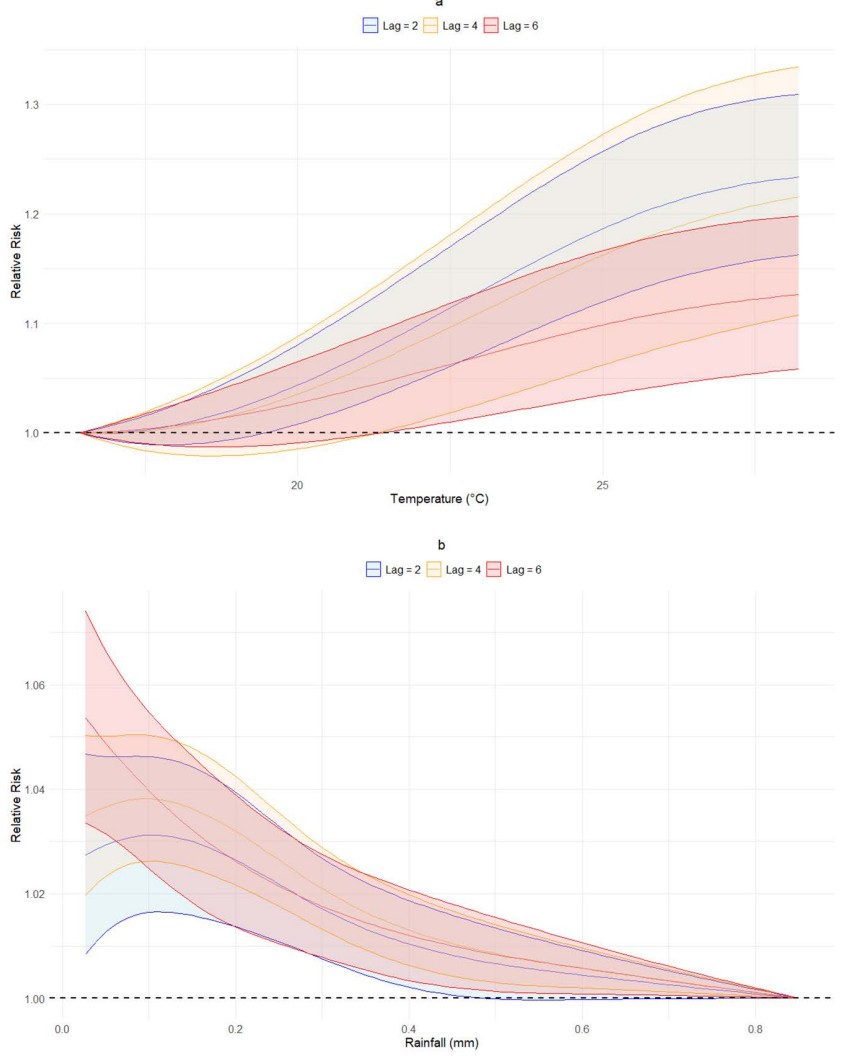

**Fig 4. Effect of temperature (a) and rainfall (b) on malaria incidence, for fixed lag values.** The dashed black horizontal line represents the null effect, i.e., where the effect is not different from that of the exposure at minimum risk (EMR).

## 4. Discussion

In this study, we estimated the lagged effects of temperature and rainfall on weekly malaria incidence in Colombia at the municipal scale for the period from 2013 to 2023. We implemented a DLNM approach to estimate the lagged effects and a Bayesian hierarchical model to account for spatiotemporal autocorrelation. Our results evidence the existence of a non-linear relationship between lagged measures of temperature and rainfall and the occurrence of malaria cases.

Our findings regarding temperature effects on malaria transmission reveal a divergence with respect to Mordecai et al. [7], who predicted optimal malaria transmission at 25 °C with dramatic declines above 28 °C. Our results at the municipal scale in Colombia showed a maximum relative risk occurring at approximately 28 °C for lag periods from 0 to 6 weeks. This higher optimal temperature aligns more closely with sporogonic development of *Plasmodium falciparum* in *Anopheles* [35] (i.e., between 24–28 °C), and with earlier vectorial capacity models that estimated transmission optima near 29 °C [36]. The discrepancy between our estimates and laboratory-derived thermal optima may reflect Colombia's distinct vector

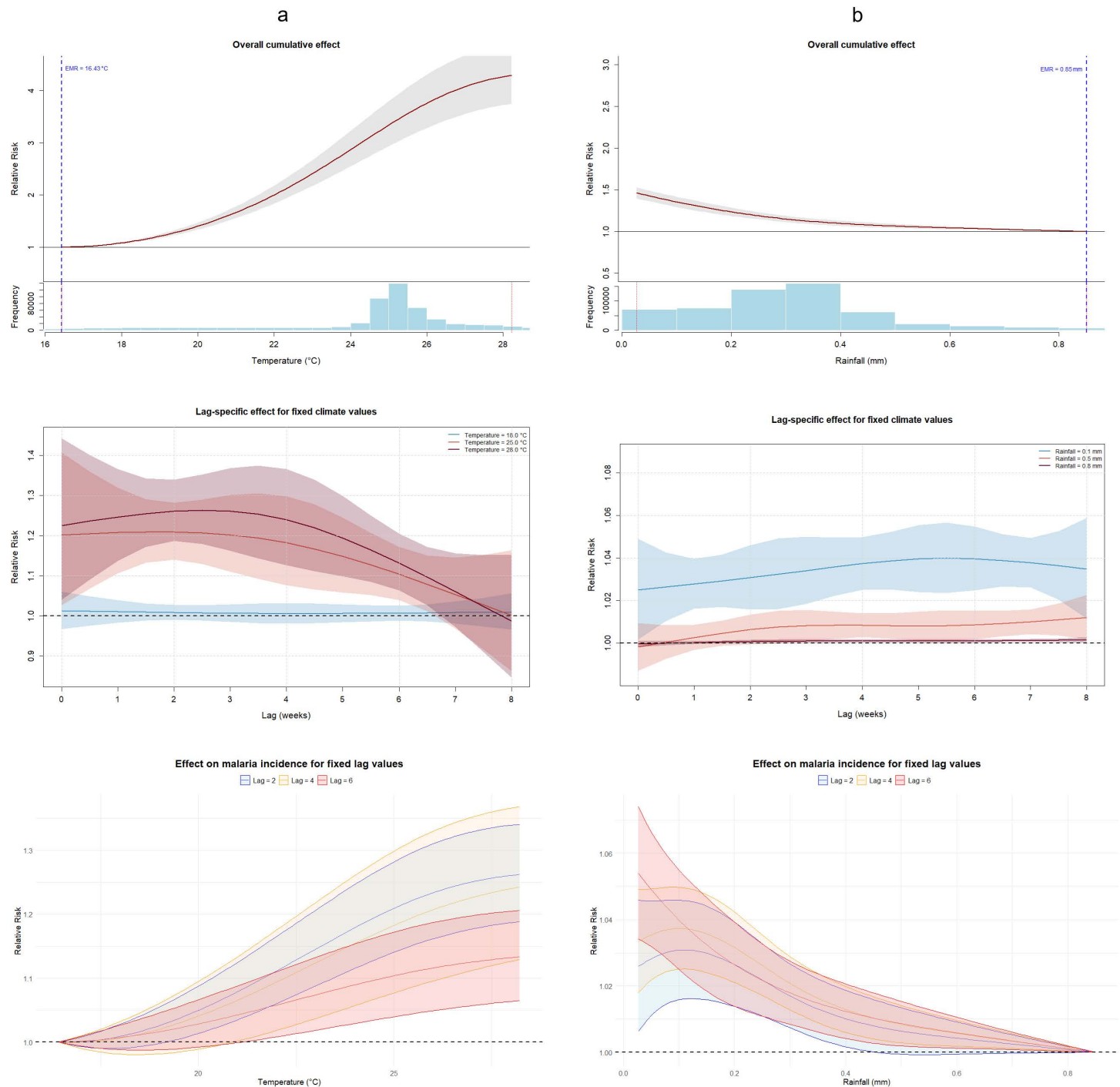

**Fig 5. Results of the sensitivity assessment of the association between temperature (a) and rainfall (b) with malaria incidence after changing the parameters of the prior and including the covariates of illegal mining and the number of holidays per week.**

species composition, microclimate variations, or behavioral adaptations of local *Anopheles* populations that extend the effective temperature range for transmission.

The inverse relationship between rainfall and malaria risk observed in our study, where lower weekly precipitation values correspond to higher transmission risk, represents a departure from previous results about malaria-rainfall dynamics. While Matsushita et al. [16] identified an optimal rainfall threshold of 120 mm monthly in Kenyan lowlands, our Colombian data suggest maximum transmission occurs during relatively drier conditions.

Studies in Mali and Zambia demonstrated that malaria cases persist throughout dry seasons at low but measurable levels [37,38], with asymptomatic parasite carriage serving as a reservoir during periods of minimal rainfall and reduced mosquito abundance. Research in Colombia documented malaria case increases following El Niño-associated dry conditions [39], consistent with our finding. The biological mechanism may involve enhanced survival of *Anopheles* larvae in persistent small water bodies that avoid washout effects [40].

Our implementation of an 8-week lag structure with weekly resolution provides temporal granularity that extends existing literature while revealing accelerated transmission dynamics in Colombian endemic zones. We observed statistically significant temperature effects primarily within 0–6 week lag periods, shorter than the 1–2 month (4–8 weeks) lags commonly reported in African highland-transition zones [12,40]. Studies in Thailand using zero-inflated Poisson DLNM documented temperature effects increasing until week 8 with humidity peaks at weeks 3–4 [22], while Ethiopian research demonstrated temperature-dependent lag variation with cases appearing 9–10 weeks after rainfall at 20 °C but only 4–5 weeks at 30 °C [40].

Our shorter effective lag periods likely reflect Colombia's consistently warmer temperatures (predominantly 22–28 °C in endemic municipalities) that accelerate both the extrinsic incubation period and mosquito gonotrophic cycles compared to cooler African highland settings. Research in Burkina Faso identified 2-month lags in moderate zones and 3-month lags in cooler regions [41], reinforcing the principle that baseline thermal regimes modulate lag structures. We hypothesize that the marked overlap in lag-specific rainfall effects across different delay periods, as observed in Fig 4b, suggests complex temporal dynamics where continuous low-level precipitation maintains transmission potential across multiple mosquito generations rather than producing discrete lagged peaks.

Our integration of spatiotemporal Bayesian hierarchical modeling with DLNM addresses a methodological void in Latin American malaria-climate research. Unlike Chinese DLNM applications that employed multilevel weekly approaches without Bayesian spatial priors [42], our INLA-DLNM synthesis explicitly models municipality-level spatial autocorrelation alongside department-level temporal random effects and time-specific structured spatial variation. This methodological approach addresses Ecuador-based research emphasizing the necessity of accounting for spatial heterogeneity in climate-malaria relationships [43].

Our analysis relies on passive surveillance data from Colombia's national health information system, which inherently carries risks of underreporting and selection bias [24]. Passive surveillance systems depend on individuals seeking care and healthcare providers correctly diagnosing and reporting cases, potentially missing asymptomatic infections and cases in remote areas with limited healthcare access [3]. Previous studies have documented that passive surveillance for malaria in Colombia may capture only a fraction of true cases, particularly during epidemic periods when health system capacity becomes overwhelmed [44]. This limitation may have led to underestimation of incidence rates and could have introduced temporal biases if reporting completeness varied systematically with climate conditions or during high-transmission periods [45].

A significant limitation of this study is the absence of data regarding the implementation of vector control campaigns and the distribution of insecticide-treated nets (ITNs) across Colombian municipalities during the study period. The lack of information on these crucial interventions represents a potential unmeasured effect modifier that could substantially influence the relationship between climate variables and malaria incidence [46,47].

Previous ecological studies examining malaria transmission have emphasized that vector control interventions, particularly ITN coverage and indoor residual spraying, can modify the climate-malaria relationship by reducing vector

populations and transmission intensity independently of meteorological conditions [47,48]. In Colombia, sporadic vector control campaigns and heterogeneous ITN distribution across endemic regions have been documented [49], yet municipality-level temporal data on intervention coverage remain unavailable in national surveillance systems.

Healthcare access barriers, particularly during extreme weather events such as heavy rainfall and flooding, can substantially hinder timely malaria diagnosis and reporting [50,51]. These barriers can lead to differential case detection rates across municipalities and time periods, introducing measurement error in our outcome variable. During periods of intense rainfall, healthcare-seeking behavior and diagnostic capacity may decline due to flooded roads, disrupted transportation, and overwhelmed health facilities [52].

Previous studies indicate that gridded climate products with resolutions similar to ERA5 (~11 km at the equator) may mask fine-scale microclimatic heterogeneity relevant to vector ecology and may not adequately capture the fine-scale microclimatic heterogeneity that influences mosquito breeding habitats and malaria transmission at local scales [53]. However, emerging evidence partially nuances this limitation; multiple mosquito species, including epidemiologically important vectors, undertake high-altitude wind-borne migrations and can travel hundreds of kilometers while carrying disseminated infections, thus remaining potentially infectious far from their point of origin [54].

This large-scale aerial connectivity suggests that, in settings where local vector populations are periodically replenished by long-distance migrants, microclimatic conditions may not fully reflect the effective environmental domain shaping vector–pathogen dynamics [54]. Consequently, while the spatial resolution of our climate data may introduce some measurement error, the capacity of mosquitoes to move across broad climatic gradients implies that the relevant exposure may operate at larger spatial scales than those defined by strictly local microclimates, thereby attenuating the expected impact of this limitation.

## 5. Conclusions

This study reveals non-linear, lagged associations between temperature and rainfall with malaria incidence across endemic municipalities in Colombia from 2013 to 2023. Contrary to previous studies, malaria risk peaked at approximately 28 °C and under relatively low rainfall conditions, with significant effects observed primarily within 0–6 weeks of exposure. These findings highlight accelerated transmission dynamics in Colombia's endemic zones, likely shaped by local vector ecology and climate regimes.

We recognize a set of limitations in our study. The reliance on passive surveillance data may underestimate true incidence and introduce reporting biases, especially during extreme weather or high-transmission periods. Additionally, the absence of municipality-level data on vector control interventions—such as insecticide-treated net distribution—and the limited spatial resolution of climate data may have introduced several forms of bias, including exposure misclassification.

Despite these constraints, our results offer actionable insights for public health authorities. By identifying specific temperature and rainfall thresholds linked to elevated malaria risk, this study supports the development of climate-informed early-warning systems. Integrating these exposure–response relationships into surveillance and intervention planning could enhance targeting of control measures, improve resource allocation, and ultimately strengthen malaria elimination efforts in Colombia.

## Supporting information

**S1 Fig. Histogram of the frequency of weekly temperature and rainfall in Colombian municipalities with altitudinal suitability for malaria transmission.**
(DOCX)

## Acknowledgments

We acknowledge the Colombian Ministry of Health for access to information on malaria cases.

## Author contributions

**Conceptualization:** Juan David Gutiérrez.

**Data curation:** Juan David Gutiérrez.

**Formal analysis:** Juan David Gutiérrez.

**Funding acquisition:** Juan David Gutiérrez.

**Investigation:** Juan David Gutiérrez.

**Methodology:** Juan David Gutiérrez.

**Project administration:** Juan David Gutiérrez.

**Resources:** Juan David Gutiérrez.

**Software:** Juan David Gutiérrez.

**Supervision:** Juan David Gutiérrez.

**Validation:** Juan David Gutiérrez.

**Visualization:** Juan David Gutiérrez.

**Writing – original draft:** Juan David Gutiérrez.

**Writing – review & editing:** Juan David Gutiérrez.

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
