## [Decision Letter · Decision Letter 0]

27 Jan 2026

PGPH-D-25-03857

Lagged Effect of Temperature and Rainfall on Malaria Incidence in Colombia (2013–2023): An Approach with Bayesian Spatiotemporal Adjustment

Dear Dr. Juan David,

Thank you for submitting your manuscript to PLOS Global Public Health. After careful consideration, we feel that it has merit but does not fully meet PLOS Global Public Health’s publication criteria as it currently stands. Therefore, we invite you to submit a revised version of the manuscript that addresses the points raised during the review process.

We look forward to receiving your revised manuscript.

Kind regards,

Srinivasa Rao Mutheneni, PhD

Academic Editor

Journal Requirements:

1. We note that you have included your Figures within the body of your manuscript. Please remove the Figures from the body of your manuscript and upload them as separate Figure files.

2. We have noticed that you have uploaded Supporting Information files, but you have not included a list of legends. Please add a full list of legends for your Supporting Information files after the references list.

Additional Editor Comments (if provided):

Reviewers' comments:

Reviewer's Responses to Questions

**Comments to the Author**

1. Does this manuscript meet PLOS Global Public Health’s publication criteria?

Reviewer #1: Partly

2. Has the statistical analysis been performed appropriately and rigorously?

Reviewer #1: I don't know

3. Have the authors made all data underlying the findings in their manuscript fully available (please refer to the Data Availability Statement at the start of the manuscript PDF file)?

Reviewer #1: No

4. Is the manuscript presented in an intelligible fashion and written in standard English?

Reviewer #1: Yes

Reviewer #1: This is a well written article on an important topic given the potential for the rise in Malaria cases due to changes in the climate. However, I have several suggestions and questions that I hope can make the article stronger:

Line 150: “We grouped the daily data by week…” Authors state that the daily temperatures data were grouped. Authors should clarify if valued were aggregated, averaged, etc.

Lines 181-184: Authors should add a note regarding the justification of their choices for the knots for temperature and rainfall (especially since the percentiles are different). Also, there should a be a note regarding the number of knots.

Lines 195-196: Authors should clarify the percentage of zeros in the data and consequently, justify the need for a zero-inflated model. At a minimum authors should report model comparison results between their model choice and a regular negative binomial model based on measures like DIC and WAIC.

Authors do not provide any measures of model fit or residual analysis to confirm the model they fitted to the data is an adequate model.

Minor comments:

Line 138: “on February 21, 2025” not necessary

**Do you want your identity to be public for this peer review?** For information about this choice, including consent withdrawal, please see our Privacy Policy

Reviewer #1: No

---

## [Decision Letter · Decision Letter 1]

24 Feb 2026

Lagged Effect of Temperature and Rainfall on Malaria Incidence in Colombia (2013–2023): An Approach with Bayesian Spatiotemporal Adjustment

PGPH-D-25-03857R1

Dear Dr Juan David Gutierrez,

We are pleased to inform you that your manuscript 'Lagged Effect of Temperature and Rainfall on Malaria Incidence in Colombia (2013–2023): An Approach with Bayesian Spatiotemporal Adjustment' has been provisionally accepted for publication in PLOS Global Public Health.

Best regards,

Srinivasa Rao Mutheneni, PhD

Academic Editor

Reviewer Comments (if any, and for reference):

Reviewer's Responses to Questions

**Comments to the Author**

Reviewer #1: All comments have been addressed

publication criteria?

Reviewer #1: Yes

3. Has the statistical analysis been performed appropriately and rigorously?

Reviewer #1: Yes

4. Have the authors made all data underlying the findings in their manuscript fully available (please refer to the Data Availability Statement at the start of the manuscript PDF file)?

Reviewer #1: Yes

5. Is the manuscript presented in an intelligible fashion and written in standard English?

Reviewer #1: Yes

Reviewer #1: Authors have satisfactorily addressed my concerns and comments.

**Do you want your identity to be public for this peer review?** For information about this choice, including consent withdrawal, please see our Privacy Policy

Reviewer #1: No
